# Differentiation Induction of Mesenchymal Stem Cells by a Au Delivery Platform

**DOI:** 10.3390/cells12141893

**Published:** 2023-07-19

**Authors:** Meng-Yin Yang, Cheng-Di Chiu, Yi-Chun Ke, Yi-Chin Yang, Kai-Bo Chang, Chien-Min Chen, Hsu-Tung Lee, Chien-Lun Tang, Bai-Shuan Liu, Huey-Shan Hung

**Affiliations:** 1Department of Neurosurgery, Neurological Institute, Taichung Veterans General Hospital, Taichung 407219, Taiwan; yangmy04@gmail.com (M.-Y.Y.); jean1007@gmail.com (Y.-C.Y.); leesd2001@gmail.com (H.-T.L.); chienluntang@gmail.com (C.-L.T.); 2Graduate Institute of Medical Sciences, National Defense Medical Center, Taipei 11490, Taiwan; 3College of Nursing, Central Taiwan University of Science and Technology, Taichung 406053, Taiwan; 4Department of Post-Baccalaureate Medicine, College of Medicine, National Chung Hsing University, Taichung 402202, Taiwan; 5Department of Neurosurgery, China Medical University Hospital, Taichung 404327, Taiwan; cdchiu4046@gmail.com; 6Spine Center, China Medical University Hospital, Taichung 404327, Taiwan; 7Graduate Institute of Biomedical Science, China Medical University, Taichung 404333, Taiwan; fish951753go@yahoo.com.tw (Y.-C.K.); kbwork2021@gmail.com (K.-B.C.); 8Division of Neurosurgery, Department of Surgery, Changhua Christian Hospital, Changhua 50006, Taiwan; 96015@cch.org.tw; 9Department of Leisure Industry Management, National Chin-Yi University of Technology, Taichung 411030, Taiwan; 10Department of Medical Imaging and Radiological Sciences, Central Taiwan University of Science and Technology, Taichung 406053, Taiwan; bsliu@ctust.edu.tw; 11Translational Medicine Research, China Medical University Hospital, Taichung 404327, Taiwan

**Keywords:** Au, stromal cell-derived factor 1α, mesenchymal stem cell, differentiation, drug delivery

## Abstract

Au decorated with type I collagen (Col) was used as a core material to cross-link with stromal cell-derived factor 1α (SDF1α) in order to investigate biological performance. The Au-based nanoparticles were subjected to physicochemical determination using scanning electron microscopy (SEM), dynamic light scattering (DLS) and ultraviolet–visible (UV-Vis) and Fourier-transform infrared spectroscopy (FTIR). Mesenchymal stem cells (MSCs) were used to evaluate the biocompatibility of this nanoparticle using the MTT assay and measuring reactive oxygen species (ROS) production. Also, the biological effects of the SDF-1α-conjugated nanoparticles (Au-Col-SDF1α) were assessed and the mechanisms were explored. Furthermore, we investigated the cell differentiation-inducing potential of these conjugated nanoparticles on MSCs toward endothelial cells, neurons, osteoblasts and adipocytes. We then ultimately explored the process of cell entry and transportation of the nanoparticles. Using a mouse animal model and retro-orbital sinus injection, we traced in vivo biodistribution to determine the biosafety of the Au-Col-SDF1α nanoparticles. In summary, our results indicate that Au-Col is a promising drug delivery system; it can be used to carry SDF1α to improve MSC therapeutic efficiency.

## 1. Introduction

Nanotechnology has been successfully applied in various clinical aspects such as minimally invasive imaging, drug delivery and engineering for tissue regeneration [1]. From tissue engineering, the applications are mainly fabricating vascular scaffolds, artificial skin, and bone tissue biomaterials [2]. The combination of nanotechnology and tissue engineering provides novel prospects for developing biological scaffolds [3] and enhancing biological performances, e.g., cell proliferation and adhesion through nanoscale surface modification techniques [4]. Thus, the use of biological nanomaterials with their outstanding biocompatibility and differentiation induction ability are powerful methods for clinical tissue engineering treatments.

Gold (denoted as Au), with sizes ranging from 1–100 nm, is a highly biocompatible metallic material [5,6]. Au can be obtained from chemical and physical manufacturing procedures. Reduction of tetrachloroauric acid by sodium citrate is used for the chemical process [7]. However, Au derived and collected from the chemical process may still possess chemical reagent residue, which can possibly lead to toxicity in biomedical approaches [8]. Therefore, several protocols have been developed to build Au, such as physical vapor deposition (PVD) and physical crushing methods [9,10]. The physical Au, when free from chemical residues, possesses outstanding purity and biocompatible features [11], which provide better characteristics for biomedical research.

Au nanoparticles can be easily fabricated into various molecules such as proteins, natural/synthetic polymers, or cell growth factors [11,12,13]. These are then very suitable for biomedical applications, including bioimaging, cancer therapy and tissue regeneration [14]. Our prior publication outlined how using gold nanoparticles modified by graphene oxide was able to enhance mesenchymal stem cell (MSC) differentiation toward neuron-like cells [15]. Additionally, the combination of electrostatic spinning fibers with Au nanoparticles was used as a vascular regeneration approach for stem cell therapies [16]. The electrostatic spinning fibers contained synthetics, e.g., poly-ethylene glycol (PEG), as well as natural polymers such as collagen and fibronectin, which are degradable and biocompatible for clinical applications, and mimic the extracellular matrix (ECM) to improve biological performances [17]. The microenvironment of the Au-based scaffolds provides better mechanical properties for MSC attachment and differentiation inducing capacities for tissue regeneration [11].

Collagen (denoted as Col) is an important protein abundant in the human body that is mainly located in connective tissues and various organs [18], and is also the major component of the ECM. Many studies have confirmed that Col offers excellent biocompatibility, biodegradability, low immunoreactivity and low cytotoxicity [18]. Collagen is a triple-helix protein composed of three polypeptide chains containing glycine, proline and hydroxyproline, which stabilize the triple helix structure [18]. Col has been widely used in biomedical research, with Type I collagen being the major one used in clinical demonstrations, while also being dominant in clinical treatments [19]. Type I Col is the major protein constituting the ECM of blood vessels, demonstrating powerful elongation and superior mechanical properties, which can resist the pressure and velocity of blood flow to avoid damage to the vascular wall [20]. Several studies have developed Type I collagen as a material for regenerative medicine. Hsieh et al. reported that Col-Au nanocomposites were able to improve the vascularization efficiency of MSCs [21]. Our prior report also indicated that Au fabricated with collagen provided the major core of a delivery system, carrying berberine and EGFR siRNA for the inhibition of breast and lung cancer cells [13,22]. This evidence shows that Au nanoparticles are an excellent candidate for use in tissue engineering and cancer treatments.

Stem cell therapy is a promising development in tissue engineering and regenerative medicine. Mesenchymal stem cells (MSCs) are pluripotent cells which preserve differentiation capacity toward various tissues through induction by specific growth factors in an appropriate microenvironment [23]. MSCs can be harvested from various tissues such as bone marrow and umbilical cord blood, and have been verified to have an excellent self-renewal ability and low immunogenicity for implantation [23]. Angiogenesis was reported to be strengthened using nanogold composites modified by fibronectin, which induce MSCs to highly express CD31 (platelet endothelial cell adhesion molecule-1) and vWF (Von Willebrand factor), which further activate the SDF-1α/CXCR4 pathway to improve cell behavior [24,25]. This Au composite has also been confirmed to induce osteoblast differentiation through upregulation of the p38/MAPK signaling pathway [26]. Moreover, after being treated with SDF-1α, the FAK/RhoA/Rac-1/Cdc42 signaling pathway was observed to activate MSCs. SDF-1α was reported to facilitate the expression of α5β3 integrin, enhancing cellular migration in tissue repair [21,24]. SDF-1α, also called chemokine 12 (CXCL12), is expressed in multiple tissues and cell types where it regulates the homing and function of many stem cells [27], including regulating chemotaxis cytokine receptor-4 (CXCR4) in hematopoietic stem or progenitor cells [28,29] and the SDF-1α/CXCR4 axis which is associated with cell motility [30].

To investigate its biocompatibility and biological performances, we used Type I collagen-decorated Au as the core vehicle for carrying SDF-1α to treat MSCs to verify is this nano delivery system can produce better effects on MSCs, particularly on their differentiation capacities for tissue regeneration.

## 2. Materials and Methods

### 2.1. Preparation of Au, Col and SDF-1α Materials

#### 2.1.1. Au Nanoparticles (Au)

A Au solution at a stock concentration of 50 ppm was obtained from Gold NanoTech Inc. (Taiwan). The Au solution was filtered through a 0.22 μm filter, which could then be considered a sterile solution. Next, 50 μL of Au solution was mixed with 950 μL of ddH_2_O or cell culture medium with the total volume becoming 1000 μL. The above mixing ratio was calculated according to the equation: M1V1 = M2V2 (M: concentration of the solution, V: volume of the solution).

#### 2.1.2. Stromal Cell-Derived Factor-1α (SDF-1α)

Stromal cell-derived factor-1α (SDF-1α) was purchased from Israel Prospec-Tany TechnoGene Ltd. (Ness-Ziona, Israel). A 100 ng/mL concentration of SDF-1α was used as a working solution in all subsequent experiments. Next, 0.5 μL of the SDF-1α stock solution was mixed with 999.5 μL of ddH_2_O or cell culture medium for a total volume of 1000 μL. The above mixing ratio was calculated according to the equation: M1V1 = M2V2 (M: concentration of the solution, V: volume of the solution).

#### 2.1.3. Type I Collagen (Col) 

A type I collagen (4.88 mg/mL) solution was purchased from BECTON DICKINSON (Franklin Lakes, NJ, USA). First, 1 mL of the type I collagen solution was mixed with 9 mL of ddH_2_O to obtain a concentration of 0.5 mg/mL Col. The above mixing ratio was calculated according to the equation: M1V1 = M2V2 (M: concentration of the solution, V: volume of the solution).

#### 2.1.4. Au Combined with Collagen (Au-Col) 

A 50 μL amount of Au solution (50 ppm) was fully mixed with 50 μL of the Col (0.5 mg/mL) solution at a 1:1 mixing ratio. Next, the above solution was mixed with 900 μL of ddH_2_O or cell culture medium to obtain a total volume of 1000 μL. The above mixing ratio was calculated according to the equation: M1V1 = M2V2 (M: concentration of the solution, V: volume of the solution).

#### 2.1.5. Au–Collagen Conjugated with Stromal Cell-Derived Factor-1α (Au-Col-SDF-1α) 

First, 50 μL of the Au solution (50 ppm) was completely mixed with 50 μL of the Col solution (0.5 mg/mL) and left standing for 8 h. Next, 0.5 μL of the SDF-1α stock solution (0.001 μg/μL) was added to the Au-Col solution. Ultimately, 899.8 μL of ddH_2_O or cell culture medium was added to obtain the Au-Col-SDF-1α solution with a total volume of 1000 μL. The above mixing ratio was calculated according to the equation: M1V1 = M2V2 (M: concentration of the solution, V: volume of the solution).

### 2.2. Characterization of As-Prepared Nanomaterials 

#### 2.2.1. Ultraviolet–Visible Spectrophotometry (UV-Vis) 

A Helios Zeta spectrophotometer (Thermo Fisher, Pittsfield, MA, USA) was applied for measurement of the UV-Vis spectra, with the wave range set from 300 to 800 nm. The as-prepared samples, including the pure Au solution, pure SDF-1α solution, Au-Col solution and Au-Col-SDF-1α solution were all subjected to examination. A quartz colorimetric tube was completely washed with deionized water prior to being wiped dry with mirror paper. First, deionized water was measured for the background absorbance. Next, each sample mentioned above was measured. Moreover, to improve the precision of data, the quartz colorimetric tube was cleaned with deionized water to remove any residues between each measurement. Origin Pro 8 (Originlab Corp., Northampton, MA, USA) software was used for data analysis, with the experiment being executed in triplicate.

#### 2.2.2. Fourier-Transform Infrared Spectroscopy (FTIR) 

The spectra of each functional group in the as-prepared nanomaterials were acquired using Fourier-transform Infrared (FTIR) spectrometry (Shimadzu FTIR Model IRPrestige-21, Tokyo, Japan) with the wavelength set 400 to 4000 cm^−1^. Next, 0.06 g of potassium bromide powder (KBr, Sigma, Burlington, MA, USA) was fully mixed with 60 μL of each as-prepared nanomaterial and dried out prior to being pressed into sheer slices. An average of 32 scans for each sample was conducted to improve the signal-to-noise percentage. The experiment was executed in triplicate.

#### 2.2.3. Scanning Electron Microscopy (SEM) 

The Au nanoparticles were observed using a scanning electron microscope (SEM, JEOL JEM-5200, JEOL Ltd., Tokyo, Japan). A silicon wafer with an addition of Au solution was dried out at 80 °C prior to sputter coating with silver. The observations were conducted under a voltage setting of 5.0 kV, with the experiment being executed in triplicate.

#### 2.2.4. Dynamic Light Scattering (DLS) Assay 

The nanomaterials, including pure Au, Au-Col and Au-Col-SDF-1α, were all subjected to dynamic light scattering (DLS) analysis. A Malvern Zetasizer Nano ZS device with a 532 nm light source at a 90° fixed scatter angle was utilized for the measurements. A test sample was prepared by adding 1 mL of each sample into a 1 cm optical path cuvette. The intensity distribution values of the samples were determined by a Zetasizer Nano ZS (Malvern Instruments, Malvern, UK) instrument with a 633 nm He-Ne laser at 25 °C and then analyzed with Malvern Zetasizer software. The experiment was executed in triplicate.

### 2.3. Biocompatibility Examinations 

#### 2.3.1. Cell Culture 

The mesenchymal stem cells used in this research were collected from the Wharton’s jelly tissue of a human umbilical cord [31]. The MSCs for the experiments were at the 8th passage, and were cautiously incubated in high-glucose Dulbecco’s Modified Eagle’s medium (H-DMEM, Invitrogen) with 10% fetal bovine serum (FBS), 1% sodium pyruvate and 1% (*v*/*v*) antibiotics (100 U/mL penicillin/streptomycin). To characterize the phenotypes of the Wharton’s jelly MSCs, the cells were detached using 2 mm EDTA with PBS. The MSCs were washed with PBS containing 2% bovine serum albumin (BSA) and 0.1% sodium azide (Sigma, USA). Next, the cells were cultured with various specific antibodies, including CD34-FITC-A, CD45-FITC-A, CD44-PE-A and CD105-PE-A (fluorescein isothiocyanate represented FITC and phycoerythrin denoted as PE). The FITC/PE-conjugated IgG1 were applied for isotype controls (BD Pharmingen, Canada) for flow cytometer detection.

#### 2.3.2. Cell Viability 

MSCs (2 × 10^4^) were loaded and seeded into a 96-well culture plate for overnight incubation to allow for cell attachment. Next, the culture medium was removed and the cells were treated with the medium (control without treatment), Au, SDF-1α, Au-Col and Au-Col-SDF-1α for 24, 48 and 72 h. After the incubation period, 100 μL of MTT (0.5 mg/mL) solution [3-(4,5-cimethylthiazol-2-yl) 2, and 5-diphenyltetrazolium bromide (Sigma-Aldrich, Burlington, MA, USA)] was added to each well and incubated for 2 h at 37 °C in a 5% CO_2_, humified atmosphere. Afterwards, the cell supernatants were removed, and 80 μL of DMSO solution (Sigma) was added for 10 min. The absorbance at 570 nm was measured with a SpectraMax M2 ELISA reader (Molecular Devices, USA). The relative fold change in cell viability was calculated according to the absorbance of treated samples compared with the untreated control group (normalized as 100%). The experiment was executed in triplicate.

#### 2.3.3. Reactive Oxygen Species (ROS) Generation 

ROS generation by the MSCs was examined using a DCFH-DA (2′,7′-dichlorofluorescin diacetate) fluorescent probe (Sigma-Aldrich, USA). MSCs (2 × 10^5^). The cells were loaded and cultured in 6-well culture plates for 24 h to allow for cell attachment. Next, the medium was removed and the cells were treated with the medium (control without treatment), Au, SDF-1α, Au-Col and Au-Col-SDF-1α for 48 h. After the above treatments, a 0.05% Trypsin–EDTA solution was applied for MSC collection and the cells were washed twice with PBS solution. The supernatant was then removed, and 10 nm of DCFH-DA solution was added in the dark for 30 min to target intracellular ROS production. The fluorescein-positive cells were detected and analyzed using a BD LSR II flow cytometer (Becton Dickinson, USA). The experiment was executed in triplicate.

### 2.4. Evaluation of Biological Performances 

#### 2.4.1. Matrix Metalloproteinases (MMPs) Activities 

The expression of MMP-2 and MMP-9 in MSCs was detected. MSCs (2 × 10^5^) were loaded and seeded into a 6-well culture plate for overnight incubation to allow for cell attachment. Next, the medium was removed, and the treatments with medium (control without treatment), Au, SDF-1α, Au-Col and Au-Col-SDF-1α were added for 48 h (37 °C). Afterwards, each culture medium containing MMP proteins was carefully harvested for gelatin zymography assay [13]. The protease digest bands in the dark blue gels could be clearly observed, while the MMPs expression was scanned and analyzed using Gel-Pro Analyzer 4.0 software (Media Cybernetics, Burlington, MA, USA). The experiment was conducted in triplicate.

#### 2.4.2. Migration Ability 

The MSC migration distance was evaluated with an Oris^TM^ migration assay kit (Platypus Technologies, Madison, WI, USA). MSCs (1 × 10^4^ cells) were incubated with Oris^TM^ seeding stoppers at 37 °C with a 5% CO_2_, humified atmosphere to achieve confluency. Afterwards, the stoppers and culture medium were removed prior to the addition of the medium (control without treatment), Au, SDF-1α, Au-Col and Au-Col-SDF-1α. After treatment for 24, 48 and 72 h in an incubator, 2 μM of a calcein-AM solution was added into each well for an another 10 min. The migratory distance was observed using a Zeiss Axio Imager A1 fluorescence microscope (Carl Zeiss AG, Jena, Germany), while it was quantitatively analyzed using Image J 5.0 software (National Institutes of Health, Bethesda, MD, USA). The experiment was executed in triplicate.

#### 2.4.3. Expression of CXCR4 and SDF-1α 

C-X-C motif chemokine receptor 4 (CXCR4) expression in the MSCs was examined through immunofluorescence staining assays. MSCs (1 × 10^4^) were cultured in 24-well culture plates overnight and then left to stand to allow for cell attachment. Afterwards, the medium was removed, and new culture medium without treatment (control), or with Au, SDF-1α, Au-Col or Au-Col-SDF-1α was added and incubated for 48 h at 37 °C. Next, the cells in each well were washed thrice with PBS and then fixed with 4% PFA (paraformaldehyde, Sigma, USA) for 30 min at 4 °C. There was then an addition of 0.5% Triton X-100 (Sigma) for 10 min to permeabilize the cells and 5% bovine serum albumin (BSA, Sigma) for blocking. Afterwards, 1.25 μg/mL of rhodamine phalloidin (Sigma) was applied to stain the cell skeleton; the cells were incubated for 30 min in the dark at room temperature (RT). Next, the MSCs were incubated with a CXCR4 primary antibody (1:500 diluted with PBS) for 8 h at 4 °C, and a 1:200 dilution of fluorescein isothiocyanate-conjugated goat anti-mouse secondary IgG antibodies for 1 h at RT. Finally, 1 μg/mL of DAPI solution (4,6-diamidion-2-phenylindole, Invitrogen, USA) was added for 10 min to target the cell nuclei. PBS solution was then applied to wash the cells three times after each step. The fluorescence images showing the cell nuclei (blue color), the cytoskeleton (red color) and CXCR4 (green color) were all captured with a Zeiss Axio Imager A1 fluorescence microscope. The expression of CXCR4 was analyzed and semi-quantified using Image J 5.0 software. The experiment was executed in triplicate.

The expression of SDF-1α in MSCs was determined through enzyme-linked immunosorbent assay (ELISA). The experiment was conducted using a Human CXCR12/SDF-1 DuoSet ELISA kit (R&D, Minneapolis, MN, USA) following the manufacturer’s instructions. MSCs (2 × 10^5^) were seeded into 6-well culture plates for overnight incubation to allow for cell attachment. Next, the medium was removed, and the new medium without treatment (control) or containing Au, SDF-1α, Au-Col, or Au-Col-SDF-1α was added and incubated for 48 h. In brief, the sample in each well was treated with primary antibodies for detection, while signal amplification was conducted through the addition of secondary HRP-conjugated antibodies. The expression of SDF-1α in each well was detected via a SpectraMax M2 ELISA reader. The experiment was executed in triplicate.

### 2.5. Measurement of Cell Progression 

#### 2.5.1. Cell Cycle Analysis 

MSCs (2 × 10^5^) were loaded into 6-well culture plates and incubated overnight to allow for cell attachment. The cells were then treated with new medium without treatment (control), or containing Au, SDF-1α, Au-Col and Au-Col-SDF-1α and incubated for 48 h. After incubation, the cells were harvested with 0.05% trypsin–EDTA (Invitrogen), washed twice with PBS, and centrifuged for 10 min (4 °C, 1200–1500 rpm). Next, the supernatant was removed, and the MSCs were then fixed using 75% alcohol (pre-cooled at −20 °C) for 8 h. Afterwards, the cells were washed twice with PBS, and 1 mL of propidium iodide (PI, Sigma) solution (containing 50 μg/mL of PI, 10 μg/mL of RNase, 10% Triton and PBS) was then added and then the cells were incubated for 30 min on ice. Cell cycle progression was analyzed using a fluorescence-activated cell sorting (FACS) Calibur flow cytometer (BD Biosciences, USA) and BD FACS Diva^TM^ software. The experiment was conducted in triplicate.

#### 2.5.2. Cell Apoptosis 

MSCs (2 × 10^5^) were loaded into 6-well culture plates and incubated overnight incubation to allow for cell attachment. The population of apoptotic cells was measured after 48 h of treatment with medium (control without treatment), Au, SDF-1α, Au-Col and Au-Col-SDF-1α through the annexin-V/PI double staining assay (Sigma-Aldrich, Burlington, MA, USA) following the manufacturer’s protocol. An Alexa Fluor 488 Annexin-V solution was applied to target phosphatidylserine (PS) on the extracellular face of the plasma membrane (representing the early cell apoptosis stage), while the cell nuclei were located using PI solution. The annexin V^+^/PI^−^ population represents early apoptotic cells, the annexin V^−^/PI^+^ population is necrotic cells and the annexin V^+^/PI^+^ population are late apoptotic and necrotic cells. Ultimately, the cells stained using the Annexin-V/PI double staining assay were detected by a FACS Calibur flow cytometer (BD Biosciences, USA) and analyzed via Flow Jo Version 7.6 software. Furthermore, the green fluorescence of the Annexin-V-positive cells was captured by a fluorescence microscope, which was further semi-quantified with Image J 5.0 software. The experiment was conducted in triplicate.

#### 2.5.3. Expression of Apoptotic Proteins via Western Blotting Assay 

MSCs (2 × 10^5^) were seeded into 10 cm^2^ culture dishes and incubated overnight to allow for cell attachment. Next, the cells were incubated with new culture medium without treatment (control), or containing Au, SDF-1α, Au-Col or Au-Col-SDF-1α for 48 h to investigate the expression of apoptosis-related proteins. The experiment was processed through the same procedure outlined in our previous study [13]. In brief, the collected cells after the above treatments were washed with PBS and incubated with a lysis buffer at 4 °C for 1 h. Next, the supernatant containing proteins was harvested through centrifugation. Afterwards, SDS-PAGE gels were prepared for the purpose of separating the protein samples, and then the samples were transferred onto PVDF membranes. The PVDF membranes were probed with a 1:1000 dilution of p21, Bax, Bcl-2, and caspase-3 primary antibodies and a 1:2000 dilution of β-actin primary antibodies (Santa Cruz, USA) for overnight incubation at 4 °C. Next, the membranes were washed with a TBST solution prior to the addition of a 1:2000 dilution of HRP-conjugated goat anti-rabbit or anti-mouse secondary IgG (Zhongshan Goldenbridge Biotechnology, Beijing, China) at RT for 60 min. An ECL kit (PerkinElmer, USA) was applied for the observation of protein bands, while protein expression was determined using a Gel-Pro Analyzer 4.0 (Media Cybernetics, USA). β-actin was used as a loading control to normalize total protein amounts and check for eventual protein degradation in the samples. The experiment was executed in triplicate.

### 2.6. Determination of Differentiation Capacities 

MSCs (1 × 10^4^) were loaded into a 24-well culture plate and incubated overnight ito allow for cell attachment. Next, the cells were treated with medium (control without treatment), Au, SDF-1α, Au-Col and Au-Col-SDF-1α for 7 days in order to investigate the expression of differentiation markers. After 7 days of incubation, the cells were washed thrice with PBS, fixed using 4% PFA for 15 min, and permeabilized by a 0.1% Triton X 100 solution prior to undergoing blocking with 5% BSA. The cells were washed three times with PBS after each step. Next, the cells were incubated with a 1:250 dilution of primary antibodies (CD31, vWF, nestin, GFAP, and β-tubulin) (Santa Cruz) at 4 °C for at least 8 h, and then washed twice with PBS solution. Afterwards, a 1:200 dilution of fluorescein isothiocyanate-conjugated goat anti-mouse/rabbit secondary IgG antibodies was added and incubated for 1 h at RT. The cell nucleus was stained with a DAPI solution for 10 min in a dark room. The green and blue fluorescence in each treatment was observed and captured by a fluorescence microscope. The green fluorescence intensity was semi-quantified with Image J 5.0 software. The experiments were conducted in triplicate.

To investigate intracellular calcium deposition, an Alizarin Red S (ARS) staining assay (Sigma) was conducted. The 1 × 10^4^ cells per well of MSCs were incubated with the various treatments for 7 days, with the procedure followed being the same as that performed in our previous research [31]. The staining images were captured with a fluorescence microscope and quantified using Image J 5.0 software. Furthermore, the adipocyte differentiation of MSCs through various treatments after 7 days of incubation was evaluated by an Oil Red O (ORO) staining assay. The protocol followed was the same as that used in our previous study [31]. The ORO-positive cells were observed through a fluorescence microscope, with the images captured from each treatment subjected to semi-quantification using Image J 5.0 software. The experiments were conducted in triplicate.

### 2.7. Real-Time Polymerase Chain Reaction (PCR) 

MSCs (2 × 10^5^) were seeded into 10 cm^2^ culture dishes and incubated overnight incubation to allow for cell attachment. Next, the cells were incubated with medium (control without treatment), Au, SDF-1α, Au-Col and Au-Col-SDF-1α for 7 days. To measure the expression of CD31, vWF, nestin, GFAP, β-tubulin, Runx-2 and PPAR, total RNA was extracted with Trizol following the procedure provided by the manufacturer. To conduct cDNA synthesis, the RevertAid First Strand cDNA Synthesis Kit (Fermentas, Canada) was used following the manufacturer’s procedures. Ultimately, the cDNA was used as templates and subjected to the IQ2 Fast qPCR System in order to process a polymerase chain reaction. The RNA expression acquired from the polymerase chain reaction was analyzed by the Step One^TM^ Plus Real-Time PCR System. The above procedures were described in our previous study [31]. The experiments were conducted in triplicate.

### 2.8. Cell Uptake Mechanisms 

#### 2.8.1. Measurement of Cellular Uptake Efficiency and Mechanisms 

MSCs (1 × 10^4^) were loaded into 24-well culture plates to allow for cell attachment after overnight incubation at 37 °C with a 5% CO_2_ atmosphere. Next, the cells were treated with FITC-labeled Au-Col and Au-Col-SDF-1α for 30 min, 2 h and 24 h. After incubating until each time point, the cells were washed thrice with PBS solution and fixed using a 4% PFA solution at 4 °C for 30 min, with 0.5% Triton X-100 (Sigma) used for permeabilization for 10 min. Afterwards, the F-actin fiber was stained with a 6 μM rhodamine phalloidin solution, while the cell nucleus was located with a 1 mg/mL DAPI solution. The cells were subjected to PBS washing prior to fluorescence observation using a Zeiss Axio Imager A1 fluorescence microscope. The green fluorescence images of Au-Col and Au-Col-SDF-1α were acquired for the quantification of uptake efficiency using Image J 5.0 software. The experiment was conducted in triplicate.

The endocytosis mechanisms in the MSCs were investigated for nanoparticle internalization. Thus, several specific inhibitors associated with cell energy-dependent endocytosis pathways were firstly prepared, including chlorpromazine (CPZ, 2 μM), methyl-β-cyclodextrin (β-MCD, 5 μM), cytochalasin D (CCD, 5 μM), and bafilomycin A (Baf, 5 μM) (Sigma). Next, 1 × 10^4^ MSCs per well were cultured in 24-well culture plates to allow for cell attachment after overnight incubation. Afterwards, the cells were pre-treated with the inhibitors for 1 h, and then subjected to incubation with FITC-labeled Au-Col and Au-Col-SDF-1α for 30 min, 2 h and 24 h. The cells, after being treated with nanoparticles, were then washed with PBS and subjected to immunofluorescence (IF) staining assay. The images containing each fluorescence were captured by a fluorescence microscope, and further subjected to fluorescence intensity quantification using Image J 5.0 software. Moreover, the fluorescein-positive cells were detected through a fluorescence activated cell sorting (FACS) Calibur flow cytometer (BD Biosciences, Canton, MA, USA), while the data were analyzed with Flow J 7.6.1. software. The experiment was conducted in triplicate.

#### 2.8.2. LysoTracker Assay 

Efficient lysosomal escape indicates the intracellular drug delivery efficiency of the polymer micelles which is associated with its biological effects. MSCs (1 × 10^4^) were seeded into 24-well culture plates and incubated overnight incubation to allow for cell attachment. The Au-Col and Au-Col-SDF-1α were labeled with FITC to further observe the fluorescence intensity. After the cells attached, they were treated with Au-Col-FITC and Au-Col-SDF-1α-FITC for 30 min, 2 h and 24 h. Afterwards, the cells were incubated with 50 nM of LysoTracker red fluorescent probe (Life Technologies, USA) for 1 h, fixed with 4% PFA for 30 min and permeabilized with 0.5% Triton X-100 for 10 min. DAPI solution was used to determine the cell nuclei location. The cells were washed thrice with PBS solution after each step. The fluorescence images were acquired using a fluorescence microscope and the fluorescence intensity was quantified with Image J 5.0 software. The experiments were executed in triplicate.

### 2.9. In Vivo Biodistribution 

Male BALB/c nude mice were provided by the National Laboratory Animal Center (Taiwan); they weighed 15–20 g and were aged 6–7 weeks. Approval for this experiment was provided by the China Medical University Animal Care and Use Committee (La-1121949). The Au-Col-SDF-1α was conjugated with FITC to further execute the retro-orbital sinus injection. Forty-eight hours after the injection, the mice were sacrificed to harvest their organs/tissues, including the brain, heart, liver, spleen, lungs and kidneys. These organs/tissues were fixed using 4% PFA, dehydrated and carefully embedded in paraffin. The integrity of each organ was evaluated through a hematoxylin and eosin (H&E, Sigma) staining histological examination, in which the tissues were cryosectioned into 4 μm thick sections. Moreover, the biodistribution of Au-Col-SDF-1α in each tissue was investigated through a fluorescence microscope. The experiments were conducted in triplicate.

### 2.10. Statistical Analysis

Statistical analysis was performed using SPSS software (version 17.0). Differences between mean values were determined by a one-way ANOVA followed by a Bonferroni’s test. Probability values (*p*) < 0.05 were regarded as a significant difference between the treatments.

## 3. Results

### 3.1. Characterization of the Physicochemical Properties

A brief illustration of the procedure regarding the synthesis of Au-Col-SDF-1α nanoparticles can be seen in Figure 1A, which were subjected to material characterization. The SEM observations of the Au nanoparticles used in the present study is shown in Figure 1B. Furthermore, Figure 1C demonstrates the histogram of the size distribution intensity determined through the DLS assay. Meanwhile, the diameters of the pure Au, Au-Col and Au-Col-SDF-1α nanoparticles were measured as 54 ± 4.5 nm, 156 ± 40.4 nm and 308 ± 33.2 nm, respectively (Figure 1D).

The UV-Vis spectrum of each as-prepared material is demonstrated in Figure 1E. There was no peak in SDF-1α due to the absence of Au nanoparticles. However, the peak at 520 nm in pure Au could be detected in Au-Col and Au-Col-SDF-1α, indicating the presence of Au nanoparticles. Additionally, the specific functional groups seen in various as-prepared materials are displayed with the FTIR spectrum (Figure 1F). The specific peaks of pure Col (pink line) at 3413 cm^−1^, 2966 cm^−1^, 1673 cm^−1^, 1556 cm^−1^ and 1377 cm^−1^, were presented as the N-H bonding, *ν*(-CH_2_), amide I, amide II and δ(-CH_2_) functional groups, respectively [32,33]. The specific bonding bands of pure Col could be found both in the Au-Col (purple line) and Au-Col-SDF-1α (brown line) nanomaterials. Furthermore, after Au-Col was combined with SDF-1α, the newly formed peak could be detected at 1186 cm^−1^, which is indicated as C-N bonding (brown line). The analyses made according to the UV-Vis and FTIR spectra demonstrated the successful manufacture of the nanomaterials.

### 3.2. Exploration of Biocompatibility and Biological Function

The phenotypes of the Wharton’s jelly MSCs used in this study were characterized by CD34, CD44, CD45 and CD105 surface markers. Appendix A demonstrates the results of each specific marker expression. The expression of the CD34 and CD45 endothelial markers was measured as 1.87% and 0.80%, respectively (negative markers). The percentage of cells expressed CD44 and CD105 markers was 99.7% and 99.6%, respectively, which demonstrates the expression of MSC surface markers by flow cytometry analysis. MSCs were subjected to the determination of cell viability by the MTT assay (Figure 2A). The MSCs were incubated with Au, SDF-1α, Au-Col and Au-Col-SDF-1α for 24, 48 and 72 h. According to the semi-quantitative results, the Au-Col-SDF-1α treatment showed the greatest cell viability (24 h: ~1.25-fold, * *p* < 0.05; 48 h: ~1.71-fold, ** *p* < 0.01; 72 h: ~2.14-fold, *** *p* < 0.001), which was significantly better than the other treatments. Meanwhile, the ROS generation in MSCs was detected with a DCFH-dA fluorescent probe through flow cytometry. The FACS histograms are organized in Appendix A, while the quantification is displayed in Figure 2B. The percentage of ROS production in MSCs was represented as 46.8% for the control, 36.4% for pure Au, 31.7% for pure SDF-1α, 25.4% for Au-Col, and 22.4% (** *p* < 0.01) for the Au-Col-SDF-1α treatment (Figure 2B). The above results indicate that Au-Col-SDF-1α significantly facilitated cell proliferation and induced the lowest ROS production in MSCs for biocompatibility identification.

The SDF-1α/CXCR4 axis is associated with the cell migratory efficiency of stem cell therapies. The CXCR4 expressed in MSCs was visualized via the IF method at 48 h with the images shown in Figure 3A. The semi-quantitative results seen in Figure 3B indicate the CXCR4 expression at 48 h of the control (1-fold), pure Au (~1.23-fold), pure SDF-1α (~1.86-fold), Au-Col (~1.85-fold) and Au-Col-SDF-1α (~2.29-fold). Furthermore, the analysis of SDF-1α expression is displayed in Figure 3C; the highest expression was induced by Au-Col-SDF-1α treatment (~1.53-fold), followed by Au-Col (~1.39-fold), SDF-1α (~1.386-fold) and pure Au (~1.01-fold).

The influence of the various treatments on the MMP activities and migratory distance of MSCs were then investigated. Figure 4A shows the MMP zymogram image, while Figure 4B shows the semi-quantified results, as analyzed based on MMP-2/9 expression intensity. The results from Figure 4B show the greatest expression of both MMP-2 (~1.87-fold, *** *p* < 0.001) and MMP-9 (~1.77-fold, ** *p* < 0.01) in the Au-Col-SDF-1α treatment when compared to other groups. Next, the green fluorescence images of migratory distance determination at 24, 48 and 72 h are exhibited in Figure 4C. To understand the detailed migratory distance, Figure 4D provides the measurement data for each treatment. The results explain that Au-Col-SDF-1α treatment also facilitated the longest migration of MSCs at each time point (24 h: 35.84 μm, 48 h: 55.97 μm and 72 h: 59.03 μm). The above evidence demonstrates that the combination of Au-Col with SDF-1α can effectively strengthen the SDF-1α/CXCR4 axis and further stimulate the expression of MMPs for MSC migration.

### 3.3. Analysis of Cell Cycle Progression and Apoptosis

To investigate the influence of Au-Col-SDF-1α on the cell cycle, the MSCs were subjected to FACS analysis. They were first treated with various nanomaterials for 48 h and then detected with flow cytometry. Figure 5A shows the histograms of each treatment labeled with markers for the SubG1, G0G1, S and G2M phases. Each population is semi-quantified and shown in Figure 5C. The results from the SubG1 phase show a ~0.46-fold decrease (***p* < 0.01) in the Au-Col-SDF-1α treatment, which means it had the lowest number of apoptotic cells when compared with the other groups (Figure 5(C-a)). There was a significant difference in the G0G1 phase between each group, as seen in Figure 5(C-b). However, based on the results seen in Figure 5(C-c,C-d), the MSC population at the S and G2M phases were highest in the Au-Col-SDF-1α treatment, with a semi-quantification of ~2.69-fold (** *p* < 0.01) and ~2.35-fold (** *p* < 0.01) increases, respectively. Additionally, the viable and apoptotic MSCs were determined through flow cytometry, with the histograms shown in Figure 5B. The viable and apoptotic cell counts were semi-quantified and are demonstrated in Figure 5(D-a,D-b). There were no significant differences in viable cell count. However, the population of apoptotic cells in the Au-Col-SDF-1α treatment was ~0.05-fold decrease (*** *p* < 0.001), which was even lower than that seen with the Au-Col treatment (~0.11-fold, ** *p* < 0.01). Subsequently, we conducted annexin-V/PI double staining assays at 48 h to evaluate the MSC apoptotic ratio, with the fluorescence images organized in Figure 6A. The green fluorescence of annexin-V-positive MSCs represents apoptosis at the early stage, which is semi-quantified in Figure 6B. The lowest apoptotic ratio was ~0.16-fold in the Au-Col-SDF-1α group, followed by ~0.29-fold in the Au-Col group, ~0.34-fold in the SDF-1α group and ~0.41-fold in the Au group. The above evidence indicates that Au-Col-SDF-1α could trigger MSCs to enter S phase for cell proliferation with the lowest induction of cell apoptosis when compared to the other treatments.

We next conducted Western blotting assays at 48 h for the evaluation of apoptosis-related protein expression in MSCs. Figure 7A shows the immunoblots of each protein with the treatments marked above the zymogram. Each protein was semi-quantified through their expression intensity, with the results exhibited as Figure 7B–F. The expression of apoptosis-induced proteins, including p21, Bax and act-caspase-3, was observed to be the lowest in the Au-Col-SDF-1α treatment, which was semi-quantified as ~0.46-fold (** *p* < 0.01), ~0.16-fold (*** *p* < 0.001) and ~0.37-fold (*** *p* < 0.001), respectively. On the contrary, the expression of anti-apoptotic protein Bcl-2 and cell cycle regulatory protein Cyclin D1 was determined to be greatest when induced by Au-Col-SDF-1α treatment, which was quantified as ~1.47-fold (** *p* < 0.01) and ~1.35-fold (** *p* < 0.01), respectively. The results show that Au-Col-SDF-1α possessed the greatest ability to inhibit cell apoptosis and enhance MSC proliferation.

### 3.4. Investigation of Differentiation Capacities

One of the major concerning issues for stem cell therapy is differentiation induction. We examined the markers of endothelial and neural differentiation via the IF method (Figure 8A–E), osteoblast differentiation through ARS staining (Figure 9A), and adipocyte differentiation through the ORO staining assay (Figure 9B) at Day 7. Each semi-quantification result was acquired based on the expression intensity. The results from Figure 8F,G demonstrate that Au-Col-SDF-1α could significantly induce endothelial differentiation within MSCs, which was quantified as ~5.32-fold (*** *p* < 0.001) for CD31 and ~5.08-fold (*** *p* < 0.001) for vWF expression. According to the quantifications seen in Figure 8H–J, MSCs treated with Au-Col-SDF-1α possessed the highest expression of neural differentiation markers, with the results quantified as ~4.75-fold (** *p* < 0.01) for nestin, ~5.05-fold (** *p* < 0.01) for GFAP and ~4.96-fold (*** *p* < 0.001) for β-tubulin expression. Moreover, Figure 9C shows that calcium deposition, revealed by ARS assay, was the best in the Au-Col-SDF-1α group (~1.70-fold, ** *p* < 0.01), while the same result is demonstrated in Figure 9D for neutral lipid accumulation (~3.70-fold, ** *p* < 0.01).

Simultaneously, we investigated the gene expression of each differentiation marker through real-time PCR, with the semi-quantitative results displayed in Figure 10A–D. The detailed data for the gene expression levels are organized in Appendix A, which demonstrates a better differentiation induction by Au-Col-SDF-1α in MSCs.

### 3.5. Examination of Cell Uptake Mechanisms

The cell uptake efficiency of Au-Col and Au-Col-SDF-1α nanoparticles by MSCs were revealed by IF staining assay at 30 min, 2 h and 24 h, with the fluorescent images exhibited in Figure 11A–C. The green fluorescence from Au-Col and Au-Col-SDF-1α was discovered to be stronger at 24 h. Semi-quantification based on the green fluorescence intensity of both nanoparticles are shown in Figure 11D,E; the results were as follows: Au-Col [30 min: 1-fold, 2 h: ~1.33-fold (*** *p* < 0.001), 24 h: ~1.40-fold (** *p* < 0.01)] and Au-Col-SDF-1α [30 min: 1-fold, 2 h: ~1.24-fold (** *p* < 0.01), 24 h: ~1.64-fold (** *p* < 0.01)].

The endocytosis pathways of MSCs to uptake Au-Col and Au-Col-SDF-1α were investigated using four types of specific endocytosis inhibitors: chlorpromazine (CPZ), methyl-β-cyclodextrin (β-MCD), cytochalasin D (CCD) and bafilomycin A (Baf). The MSCs were pre-treated with the specific inhibitors, and then treated with Au-Col and Au-Col-SDF-1α for 30 min, 2 h and 24 h. The Au-Col uptake images are displayed in Figure 12A, while the semi-quantitative results were obtained through IF (Figure 12C) and FACS (Figure 12D) methods. Moreover, the fluorescence results of Au-Col-SDF-1α uptake efficiency at each time point are also shown in Figure 12B, with the IF and FACS quantifications organized in Figure 12E,F. According to the demonstration seen in the uptake images, the green fluorescence intensity of Au-Col was found to be weak in the CPZ and Baf groups (Figure 12A): 30 min [CPZ (~0.59-fold) and Baf (~0.92-fold)]; 2 h [CPZ (~0.55-fold, * *p* < 0.05) and Baf (~0.47-fold, ** *p* < 0.01)]; 24 h [CPZ (~0.42-fold, ** *p* < 0.01) and Baf (~0.38-fold, ***p* < 0.01)] (Figure 12C). The FACS results in Figure 12D are 30 min [CPZ (~1.01-fold) and Baf (~0.73-fold)]; 2 h [CPZ (~0.67-fold, ** *p* < 0.01) and Baf (~0.59-fold, ** *p* < 0.01)]; 24 h [CPZ (~0.27-fold, *** *p* < 0.001) and Baf (~0.31-fold, *** *p* < 0.001)]. Furthermore, a similar trend was also found in the analysis of Au-Col-SDF-1α uptake efficiency (Figure 12B): 30 min [CPZ (~0.32-fold, ** *p* < 0.01) and Baf (~0.41-fold, * *p* < 0.05)]; 2 h [CPZ (~0.45-fold, ** *p* < 0.01) and Baf (~0.44-fold, ** *p* < 0.01)]; 24 h [CPZ (~0.55-fold, * *p* < 0.05) and Baf (~0.43-fold, ** *p* < 0.01)] (Figure 12E). The FACS results seen in Figure 12F are 30 min [CPZ (~0.87-fold) and Baf (~1.16-fold, * *p* < 0.05)]; 2 h [CPZ (~0.64-fold) and Baf (~0.88-fold)]; 24 h [CPZ (~0.31-fold, *** *p* < 0.001) and Baf (~0.27-fold, *** *p* < 0.001)]. The histograms of fluorescein-positive MSCs detected by flow cytometry at each time point are exhibited in Appendix A. According to the above evidence, we determined that the uptake of both Au-Col and Au-Col-SDF-1α by MSCs was significantly inhibited by CPZ and Baf endocytosis inhibitors, which are associated with clathrin-mediated endocytosis and the vascuolar-type H^+^-ATPase pathway.

The intracellular transportation in MSCs to uptake Au-Col and Au-Col-SDF-1α was investigated using LysoTracker staining assay at 30 min, 2 h and 24 h. The images of both nanoparticles are organized in Figure 13A–C, while the semi-quantitative results are displayed in Figure 13D,E. The result of Au-Col fluorescence intensity was 30 min: 1-fold; 2 h: ~1.07-fold (* *p* < 0.05); and 24 h: ~1.43-fold (*** *p* < 0.001) (Figure 13D). The data for Au-Col-SDF-1α fluorescence intensity were 30 min: 1-fold; 2 h: ~1.33-fold (*** *p* < 0.001); and 24 h: ~1.40-fold (*** *p* < 0.001) (Figure 13E). The above evidence demonstrates that Au-Col-SDF-1α could be significantly absorbed by MSCs, and that their outstanding stability allows Au-Col-SDF-1α nanoparticles to escape metabolism by lysosomes in cells.

### 3.6. Assessments of Biodistribution in an Animal Model

Au-Col-SDF-1α was labeled with FITC in order to investigate the organ/tissue integrity and biodistribution in an animal model. After retro-orbital sinus injection, the time point of 24 h was chosen for to acquire organs, including the brain, heart, liver, spleen, lungs and kidneys. According to the H&E staining images displayed in Figure 14A,B, the injection of Au-Col-SDF-1α did not lead to serious tissue damage in any of the organs. Afterwards, the particle biodistribution determined by fluorescence images, shown in Figure 15A,B, indicated that the FITC-labeled Au-Col-SDF-1α could be observed in most tissues. In summary, Au-Col-SDF-1α may be a potential nanodrug in stem cell therapies owing to its better induction of differentiation and greater in vivo retention efficacy. The schematic diagram is illustrated in Figure 16.

## 4. Discussion

The use of a nanodrug carrier is the current method for drug release as it can allow the nanodrug to be transported to specific tissues/organs for effective regeneration of the human body. To decrease the degradation rates in the transportation process, the drug is encapsulated with natural polymers. The drug is slowly released within the fluids of the human body to achieve superior therapeutic effects [34]. The stromal cell-derived factor 1 (SDF-1) can guide the migration of hematopoietic cells from the liver to bone marrow for the enhancement of angiogenesis in embryonic development, and is associated with tumor cell metastasis and cell motility [35]. Furthermore, the stromal cell-derived factor 1α (SDF-1α), also known as CXCL12, is a chemokine protein translated from the CXCL12 gene. CXCR4 (C-X-C chemokine receptor type 4), also called fusin or CD184, is a 7-transmembrane (7TM) protein translated from the CXCL12 gene [36]. SDF-1α and CXCR4 can be classified as ligand and receptor, respectively. When the binding of both proteins occurs, the cell signal will activate to influence cell survival, proliferation and migration ability [37]. Therefore, to develop the nano delivery platform with superior biocompatibility and induction of differentiation capacities, SDF-1α was carried by the Au-Col nano platform and used to treat MSCs for various assessments, including biocompatibility, biological performances and differentiation.

The available literature suggests that mesenchymal stem cell therapy can enhance regenerative efficiency through secreting growth factors in specific damaged tissues, with the stimulating of cell differentiation factors occurring to facilitate tissue repair [23]. Furthermore, with the advantages of low inflammation and immunogenic induction, MSCs can be recruited in high numbers to the injured sites [38]. Physical Au is a highly biocompatible nanometallic particle which has been applied to crosslink with bioactive molecules in order to regulate cell motility and differentiation [39]. Our previous research has demonstrated that pullulan–collagen–Au nanocomposites could reduce inflammation responses and induce MSCs to undergo neural differentiation via the expression of nestin, GFAP and β-tubulin. As previously mentioned, Col is a natural polymer which provides superior mechanical properties for vascular scaffolds [18]. Thus, based on the results shown in Figure 2, Au-Col-SDF-1α facilitated the highest cell proliferation rate with the lowest production of intracellular ROS within MSCs. Figure 5, Figure 6 and Figure 7 in the present study show the cell cycle progression and apoptosis evaluation results. Normal cell cycle progress can be categorized into the G0G1, S and G2M phases. The G1 phase is the regulator of cell growth. During the S phase, DNA replication occurs to prepare for cell proliferation; when the DNA replication is complete, the G2 phase is initiated to prepare for cell mitosis. Finally, the M phase represents cell mitosis for living cells. However, the cells that cannot enter into the normal cell cycle will undergo the Sub G1 phase, which is cell apoptosis [40]. In Figure 5C, the number of MSCs in the Sub G1 phase were lowest in the Au-Col-SDF-1α treatment, while the cell populations in the S and G2M phases were the greatest. The results indicate that Au-Col-SDF-1α could significantly induce MSCs to proliferate and avoid apoptosis. Furthermore, during cell apoptosis, the annexin-V probe can target the phosphatidylserine (PS) on the cell membrane, which is another marker of apoptosis [41]. The FACS and IF results from Figure 5D and Figure 6 demonstrate the decreased ratio of apoptotic cells in each treatment. However, the Au-Col-SDF-1α-treated MSCs possessed the lowest apoptotic ratio. Additionally, Figure 7 shows the relationship between cell cycle and apoptosis-related protein expression. cyclin D1 is a protein that regulates the cell cycle [42], while the p21 protein antagonizes cyclin D1 [43]. Bax-2 and caspase-3 are considered to be proteins which cause nuclear damage [44]. On the contrary, Bcl-2 is an anti-apoptosis factor [45]. The evidence has demonstrated that Au-Col-SDF-1α can significantly induce MSCs to express cyclin D1 and Bcl-2 to strengthen cell growth and proliferation, indicating that Au-Col-SDF-1α is an outstanding, biocompatible, nano delivery platform.

MSCs could be induced by various specific factors to undergo differentiation with better proliferation abilities [23]. Various markers, including those marking endothelial, neural, osteogenic and adipogenic differentiation have been identified. The angiogenic factors, including CD31 and vWF, were associated with angiogenesis for vascular regeneration. When discussing neural differentiation markers, nestin is capable of representing differentiation into glia and keratinocyte [46], GFAP (glial fibrillary acidic protein) can verify astrocyte differentiation [47], and β-tubulin is the protein component of microtubules in neural cells [48]. In line with the assessments of differentiation capacities in this study, we discovered that the expression of endothelial, neural, osteogenic and adipogenic markers was remarkably upregulated by the addition of Au-Col-SDF-1α. This is the potential testimony showing that Au-Col-SDF-1α can effectively influence MSC differentiation. SDF-1α/CXCL12-CXCR4 engagement on DCs is known to promote DC maturation, survival and migration. In our prior report, we conjugated SDF-1α into an Au–collagen nanocarrier and detected its biological effect on dendritic cells [49]. This may provide a promising approach for the specific targeting of immunotherapy applications. Therefore, compared with this study, Au-Col-SDF-1α not only stimulated stem cell differentiation by activating the SDF-1α/CXCR4 signaling pathway, but it also activated T cells by stimulating dendritic cells, which can be used for immunotherapy, such as the development of novel gold nanocomposite vaccines.

The endocytosis mechanisms of MSCs to uptake Au-Col and Au-Col-SDF-1α were then discussed. The endocytosis processes of cells can be classified as clathrin-mediated endocytosis, caveolae-mediated endocytosis, micropinocytosis and phagocytosis [50,51]. To clarify the detailed pathways, we selected four specific inhibitors: 1. chlorpromazine (CPZ), 2. methyl-β-cyclodextrin (β-MCD), 3. cytochalasin D (CCD) and 4. bafilomycin A (Baf). According to published studies, CPZ is an inhibitor of clathrin-mediated endocytosis [52]; β-MCD depresses caveolae-mediated endocytosis [53]; and CCD can cause the depolymerization of actin. Therefore, CCD can block both macropinocytosis and phagocytosis [54] and Baf can decrease vascuolar-type H^+^-ATPase pathway activity, which is associated with clathrin-mediated endocytosis [55]. Figure 12 demonstrates that both Au-Col and Au-Col-SDF-1α nanoparticles were taken up by MSCs via clathrin-mediated endocytosis and vascuolar-type H^+^-ATPase pathways. Additionally, the stability of Au-Col-SDF-1α in MSCs was discussed based on Figure 13. Lysosomes are membrane-bound organelles which are related to endocytosis. The pH value in the lysosomes is approximately 4.5 to 5.0, which is an acidic environment to recruit hydrolytic enzymes to degrade foreign substances [56]. The bio-distribution of Au-Col was studied 24 h after an injection into the retro-orbital sinus. The histological analysis showed that the FITC-labeled Au-Col could be seen in most organs after injection, including the heart, spleen, brain, lungs, kidneys and liver, as was observed in our previous study [22]. Another report also demonstrated that, when administered into mice via retro-orbital sinus injection, Au-Col nanocarriers were also found to be highly expressed in different organs, which displayed of strong tissue integrity. Based on these findings, a Au-Col nanocarrier may be safe and offer a better retention efficacy in the treatment of mice, which suggests it may have potential in the development of nanodrugs for the treatment of either breast or brain tumors [37,48]. However, additional experiments must be conducted in the future in order to address the targeting efficiency issues of Au for drug or biomolecule delivery. For example, the pharmacokinetics and biodistribution of Au-Col-SDF-1α should both be taken into account. Indeed, a robust standardized assay platform must be set up for the assessment Au-Col-SDF-1α in vivo for the elucidation of long-term retention time, while interactions with the immune microenvironment must be investigated through additional studies in the future. The evidence demonstrate that both Au-Col and Au-Col-SDF-1α would not be degraded by lysosomes in an acidic environment, and therefore Au-Col-SDF-1α can act as a stable drug delivery platform.

## 5. Conclusions

In the present study, we manufactured a Au-Col nano delivery system to carry SDF-1α for the investigation of its biocompatibility and biological performance, particularly in inducing multilineage differentiation of MSCs. The increased CXCR4 and SDF-1α expression in the MSCs proved that Au-Col-SDF-1α could significantly strengthen the SDF-1α/CXCR4 pathway, with an enhancement in MMP activities and migratory ability. Furthermore, Au-Col-SDF1α facilitated MSC differentiation, including endothelial, neural, osteoblast and adipocyte differentiation. Moreover, cell cycle progression and apoptosis analysis show the induction of cell proliferation and inhibition of cell death. In conclusion, the evidence of solid biological performance with biocompatibility demonstrates that Au-Col-SDF-1α is a promising candidate, with further exploration of the endocytosis mechanisms possibly providing the directions needed for future clinical stem cell therapies.

## Figures and Tables

**Figure 1 cells-12-01893-f001:**
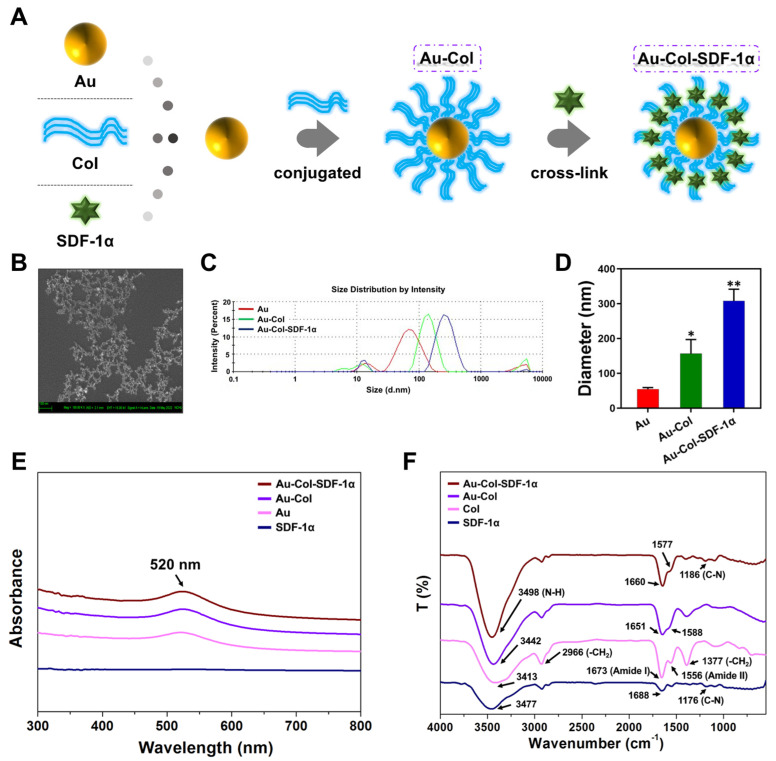
Characterization of Au-derived nanoparticles. (**A**) Brief procedure for the preparation of Au-Col and Au-Col-SDF-1α nanoparticles. (**B**) The SEM images for the observation of Au nanoparticles. (**C**) The size distribution intensity determined by DLS assay. (**D**) The diameter of Au, Au-Col, and Au-Col-SDF-1α was detected. * *p* < 0.05, ** *p* < 0.01: compared to the Au group. (**E**) The UV-Vis spectra, which indicate the specific absorbance peak at 520 nm for the Au nanoparticles. (**F**) The FTIR spectra indicate the functional groups in each nanomaterial. All the results exhibited are from one of three independent experiments.

**Figure 2 cells-12-01893-f002:**
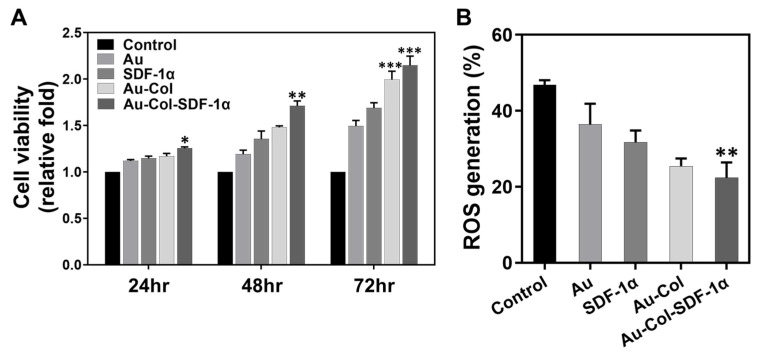
Biocompatibility assessments of various treatments within MSCs. (**A**) The cell viability of MSCs was evaluated by MTT assay at 24, 48 and 72 h. The results indicate that Au-Col-SDF-1α treatment significantly enhanced MSC proliferation. (**B**) The intracellular ROS generation in MSCs induced by different treatments was quantified at 48 h. The results show that Au-Col-SDF-1α treatment triggered the lowest ROS production in MSCs. The above results were quantified in triplicate. * *p* < 0.05, ** *p* < 0.01, *** *p* < 0.001: compared to the control group.

**Figure 3 cells-12-01893-f003:**
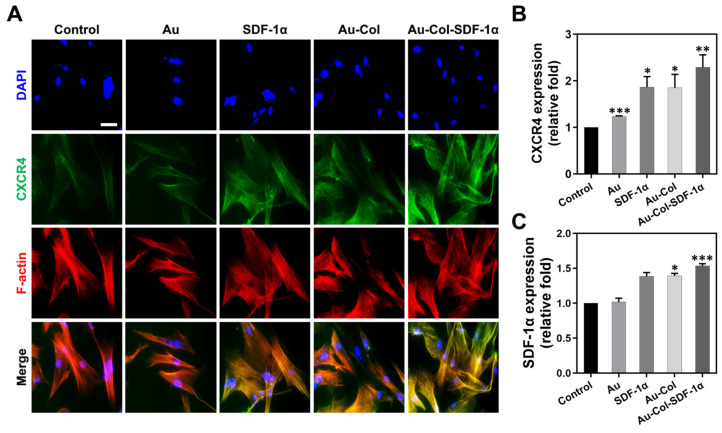
CXCR4 and SDF-1α expression in MSCs at 48 h after treatment. (**A**) The fluorescent images of CXCR4 expression are displayed. Blue color: cell nucleus stained by DAPI; green color: CXCR4 in MSC; red color: F-actin fibers. The images presented are from one of three independent experiments. Scale bars = 20 μm. (**B**) The CXCR4 expression amount was semi-quantified based on the fluorescence intensity. (**C**) The semi-quantification of SDF-1α expression in MSCs was determined through ELISA assay. The above results were quantified in triplicate. * *p* < 0.05, ** *p* < 0.01, *** *p* < 0.001: compared to the control group.

**Figure 4 cells-12-01893-f004:**
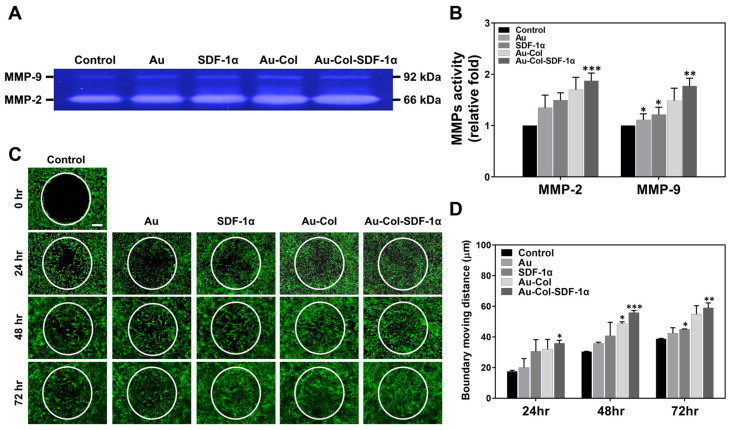
MMPs activities and migratory ability of MSCs. (**A**,**B**) The zymogram of MMP-2 and MMP-9 expression after 48 h of treatment is displayed, while the expression intensity was semi-quantified. (**C**,**D**) The boundary moving distance of MSCs induced by various treatments were captured by a fluorescence microscope at 24, 48 and 72 h. The image at 0 h represents the control reference. Calcein-AM was used for staining living cells (green color). The migratory distance was measured at each time point. Scale bars = 100 μm. The above results were quantified in triplicate. * *p* < 0.05, ** *p* < 0.01, *** *p* < 0.001: compared to the control group.

**Figure 5 cells-12-01893-f005:**
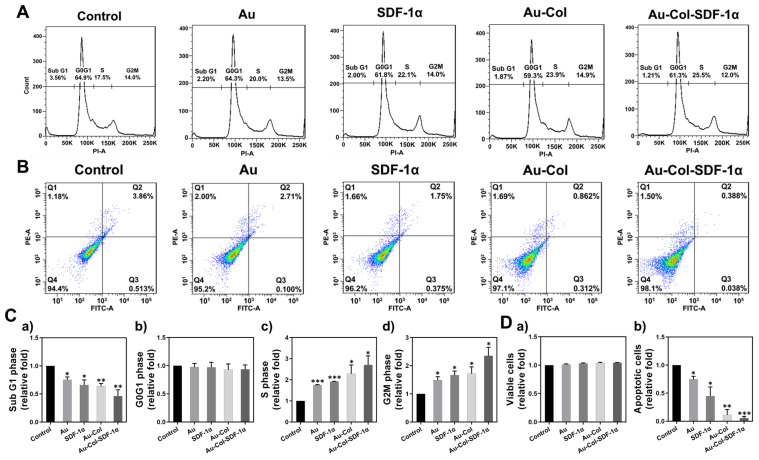
Population of MSCs in cell cycle progression and apoptotic cell death after various treatments for 48 h. (**A**) The flow histograms of MSC cell cycle analysis. (**B**) The flow histograms of the population of apoptotic cells are exhibited. (**C**) The population of MSCs in the SubG1, G0G1, S and G2M phases under various treatments were semi-quantified. (**D**) The semi-quantitative results of viable and apoptotic MSCs are shown. The histograms are the data from one of three independent experiments. The above results were quantified in triplicate. * *p* < 0.05, ** *p* < 0.01, *** *p* < 0.001: compared to the control group.

**Figure 6 cells-12-01893-f006:**
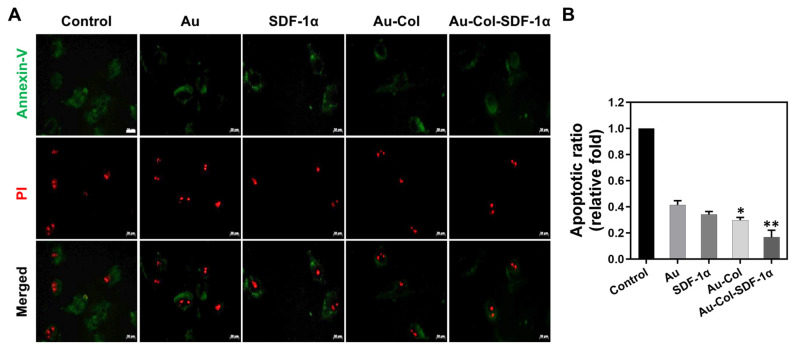
Apoptotic MSCs induced by various treatments examined through Annexin-V/PI double staining assay at 48 h. (**A**) The fluorescence images of MSCs are displayed; the green color indicates apoptotic cells and the red color indicates the cell nucleus. The images are from one of three independent experiments. Scale bars = 20 μm. (**B**) The green fluorescence intensity was measured for the semi-quantifications. The results were quantified in triplicate. * *p* < 0.05, ** *p* < 0.01: compared to the control group.

**Figure 7 cells-12-01893-f007:**
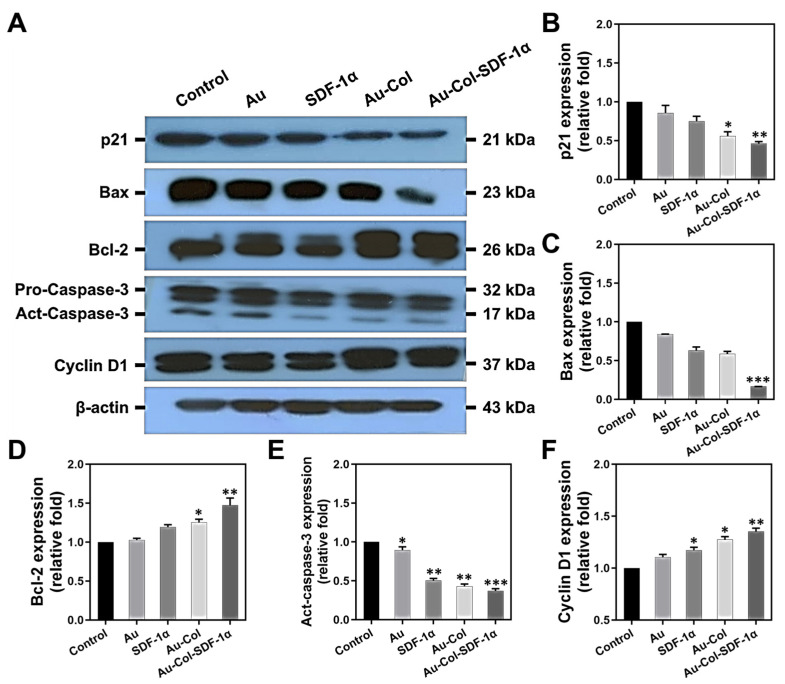
Expression of apoptosis-related proteins in MSCs stimulated by different treatments at 48 h. (**A**) The zymograms of protein immunoblots are demonstrated by Western blotting assay. The semi-quantitative results of each protein including (**B**) p21, (**C**) Bac, (**D**) Bcl-2, (**E**) Act-caspase-3, and (**F**) cyclin D1 are shown. The above results were quantified in triplicate. * *p* < 0.05, ** *p* < 0.01, *** *p* < 0.001: compared to the control group.

**Figure 8 cells-12-01893-f008:**
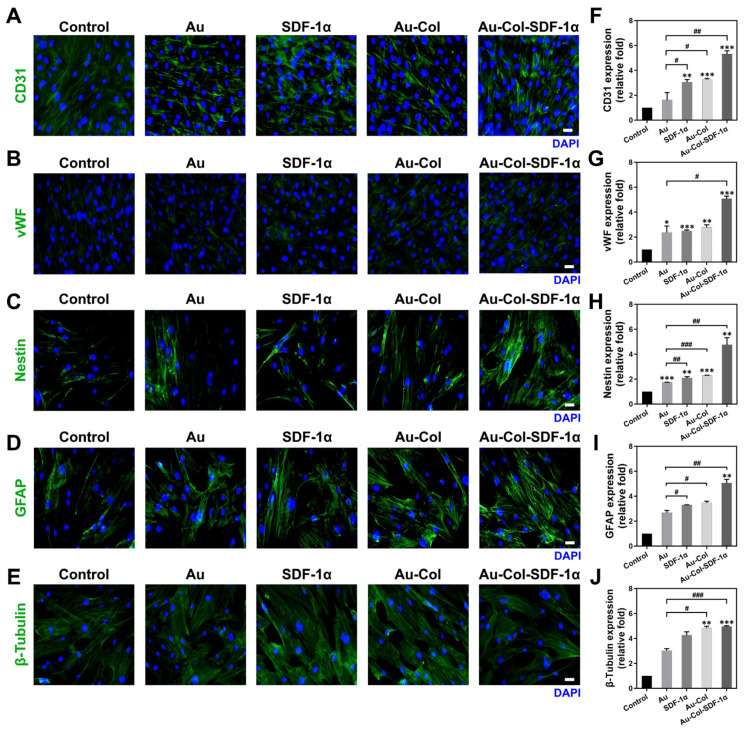
Differentiation capacities of MSCs induced by various treatments measured at Day 7. The expression differentiation markers including (**A**) CD31, (**B**) vWF, (**C**) nestin, (**D**) GFAP, and (**E**) β-tubulin were detected using the immunofluorescence method. Green color indicates the expression of each marker, while the blue color is the cell nuclei located by DAPI. Scale bars = 20 μm. (**F**–**J**) The semi-quantification of each marker was determined based on the fluorescence intensity. The above results were quantified in triplicate. * *p* < 0.05, ** *p* < 0.01, *** *p* < 0.001: compared to the control group; ^#^
*p* < 0.05, ^##^
*p* < 0.01, ^###^
*p* < 0.001: compared to the pure Au group.

**Figure 9 cells-12-01893-f009:**
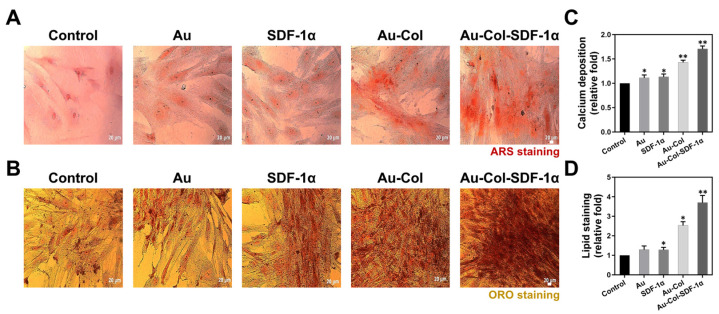
Investigation of osteoblast and adipocyte differentiation of MSCs after various treatments at Day 7. (**A**) The images of calcium deposition in MSCs were acquired using ARS staining assay. (**B**) The images of neutral lipids in MSCs were obtained from ORO staining assay. Scale bar = 20 μm. The semi-quantitative results for (**C**) calcium accumulation and (**D**) neutral lipid content are shown. The above results were quantified in triplicate. * *p* < 0.05, ** *p* < 0.01: compared to the control group.

**Figure 10 cells-12-01893-f010:**
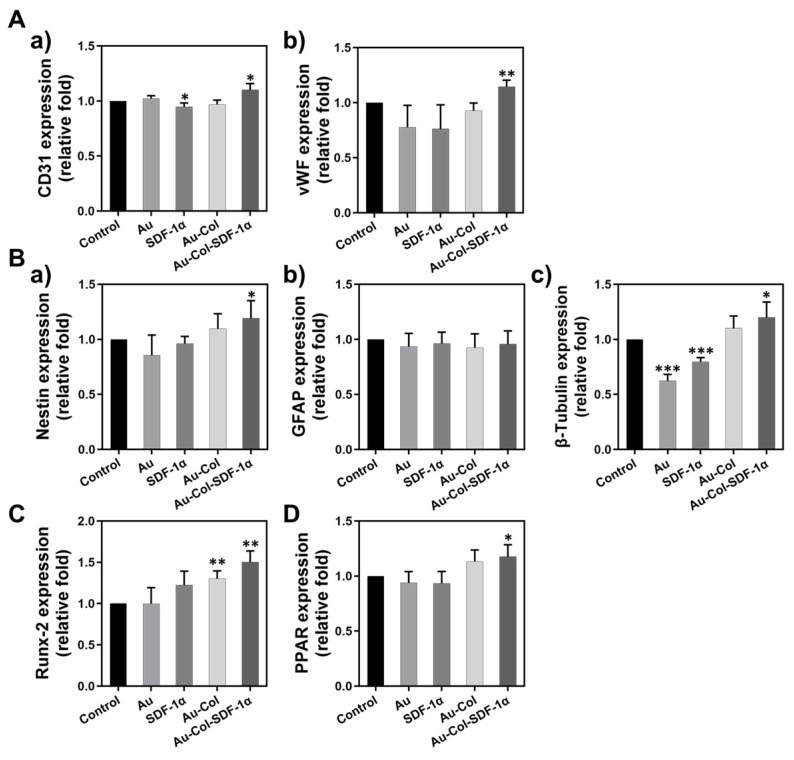
Gene expression of the various differentiation markers in MSCs was determined with real-time PCR assay at Day 7. (**A**-**a**,**A**-**b**) The expression of CD31 and vWF endothelial differentiation markers were semi-quantified. (**B**-**a**–**B-c**) The expression of the nestin, GFAP and β-tubulin neural differentiation markers were semi-quantified. (**C**) The expression of the Runx-2 osteoblast differentiation marker was semi-quantified. (**D**) The expression of the PPAR adipocyte differentiation marker was semi-quantified. The above results were quantified in triplicate. * *p* < 0.05, ** *p* < 0.01, *** *p* < 0.001: compared to the control group.

**Figure 11 cells-12-01893-f011:**
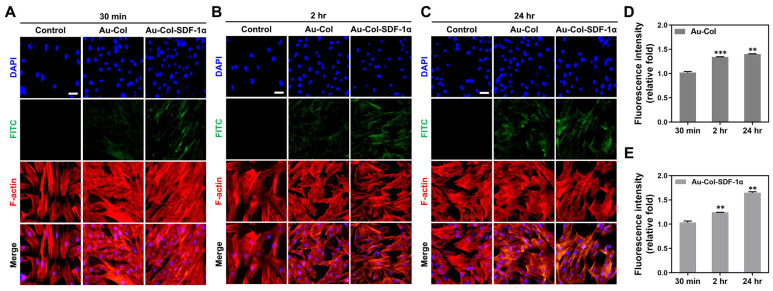
Evaluation of cell uptake efficiency by MSCs. The FITC-labeled Au-Col and Au-Col-SDF-1α were subjected to MSC uptake investigations at (**A**) 30 min, (**B**) 2 h and (**C**) 24 h. The green fluorescence images indicate the presence of Au-Col and Au-Col-SDF-1α in cells. Scale bar = 20 μm. Moreover, the semi-quantifications based on fluorescence intensity at each time point are shown as (**D**) Au-Col and (**E**) Au-Col-SDF-1α treatments. The results were quantified in triplicate. ** *p* < 0.01, *** *p* < 0.001: compared to the 30 min group.

**Figure 12 cells-12-01893-f012:**
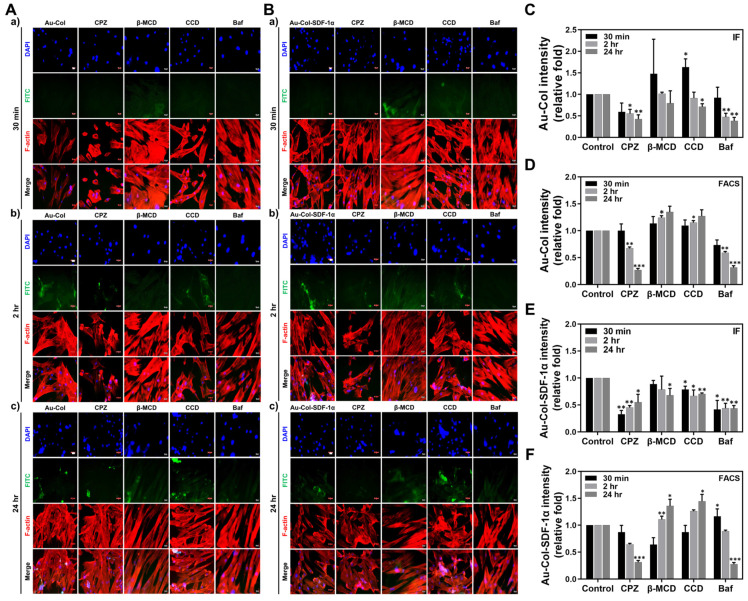
Endocytosis mechanisms of MSCs investigated through various specific inhibitors at 30 min, 2 h and 24 h. (**A**) The fluorescence images of MSC uptake of Au-Col are exhibited. (**B**) The fluorescence images of MSC uptake of Au-Col-SDF-1α are completely demonstrated. Scale bar = 20 μm. (**C**,**D**) The quantitative results of Au-Col uptake amounts in MSCs based on green fluorescence intensity were determined through the IF and FACS methods. (**E**,**F**) Furthermore, the results for Au-Col-SDF-1α uptake by MSCs were also quantified through IF and FACS assays. The quantifications were measured in triplicate. * *p* < 0.05, ** *p* < 0.01, *** *p* < 0.001: compared to the 30 min group.

**Figure 13 cells-12-01893-f013:**
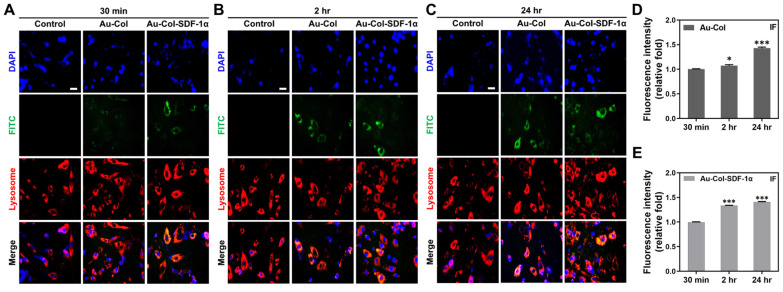
Internalization of Au-Col and Au-Col-SDF-1α by MSCs was detected by LysoTracker staining assay. The fluorescence images of the cell nucleus (blue color), Au-Col and Au-Col-SDF-1α nanoparticles (green color), lysosomes (red color) are shown at (**A**) 30 min, (**B**) 2 h and (**C**) 24 h. Scale bar = 20 μm. (**D**) The semi-quantification of Au-Col green fluorescence intensity was measured. (**E**) The uptake of Au-Col-SDF-1α into MSCs were quantified based on the fluorescence intensity. The data were measured in triplicate. * *p* < 0.05, *** *p* < 0.001: compared to the 30 min group.

**Figure 14 cells-12-01893-f014:**
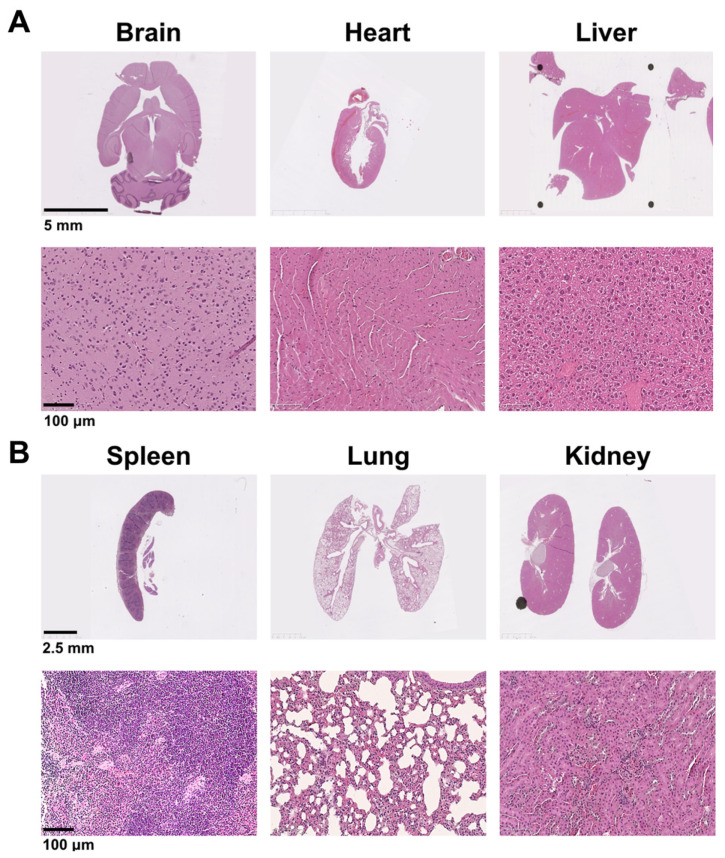
Investigation of tissue integrity after 24 h of Au-Col-SDF-1α treatment. The FITC-labeled Au-Col-SDF-1α was injected into the mice through retro-orbital sinus injection. The tissue integrity of the brain, heart, liver, spleen, lungs and kidneys were evaluated by H&E staining assay. (**A**) The tissue morphology of the brain, heart and liver was observed. Scale bar = 5 mm and 100 μm. (**B**) The tissue morphology of the spleen, lungs and kidneys was revealed. Scale bars = 2.5 mm and 100 μm.

**Figure 15 cells-12-01893-f015:**
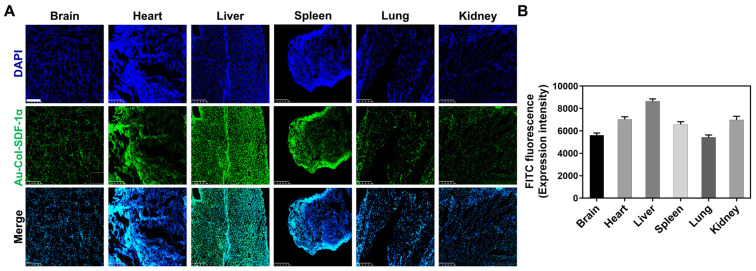
Exploration of particle biodistribution after 24 h of treatment. Green fluorescence of Au-Col-SDF-1α in the brain, heart, liver, spleen, lungs and kidneys was observed through a fluorescence microscope. (**A**) The images of each tissue are displayed. Scale bars = 250 μm. (**B**) The green fluorescence intensity was quantified, and the results indicate that Au-Col-SDF-1α was present in most tissues. The cell nucleus was visualized by DAPI.

**Figure 16 cells-12-01893-f016:**
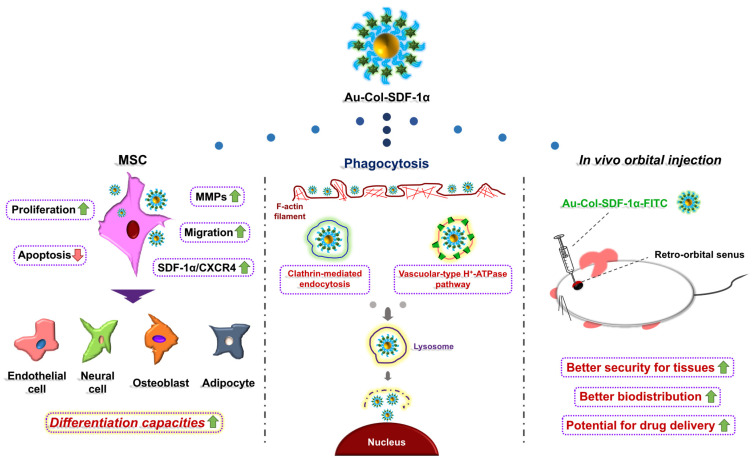
Schematic illustrations for the efficiency of Au-Col nanoparticles to carry SDF-1α. The biological performances, such as differentiation capacities in MSCs can be enhanced by Au-Col-SDF-1α, while the endocytosis pathways of MSCs to absorb nanoparticles have also been explored. Furthermore, the orbital injection in an animal model demonstrated that the nanoparticles have high stability and better biodistribution. In summary, Au-Col nanoparticles carrying SDF-1α combined with stem cell therapy can be a potential nanodrug delivery system.

## Data Availability

All data are contained within the article.

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
