# Peer review of "Differentiation Induction of Mesenchymal Stem Cells by a Au Delivery Platform"

_cells, 2023, doi:10.3390/cells12141893_

Round 1

Reviewer 1 Report

Yang et al. the investigated biocompatibility and biological performances, manufacturing Au nanoparticles with Type I collagen as the core system for carrying SDF-1α, with the MSCs, expecting that this Au-Col-SDF-1α nano delivery platform can stimulate better performances in MSCs, particularly in the differentiation capacities for regenerative tissue medicine.

This a very interesting study with an adequate experimental methodology to support the data.

This reviewer would like to suggest some improvements and clarifications to achieve the amelioration of the final article. See the comments below:

1 -) Please stated the limitation of the study.

Author Response

Reviewer 1:

Yang et al. the investigated biocompatibility and biological performances, manufacturing Au nanoparticles with Type I collagen as the core system for carrying SDF-1α, with the MSCs, expecting that this Au-Col-SDF-1α nano delivery platform can stimulate better performances in MSCs, particularly in the differentiation capacities for regenerative tissue medicine.

This a very interesting study with an adequate experimental methodology to support the data.

This reviewer would like to suggest some improvements and clarifications to achieve the amelioration of the final article. See the comments below:

1 -) Please stated the limitation of the study.

Response:

Thanks the valuable comment from the reviewer. We have included the description of limitation in the “Discussion” section “However, additional experimentations must be conducted to address targeting efficiency issues of gold nanoparticle for drug or biomolecules delivery from this article in the further. For example, taking into account that pharmacokinetics and biodistribution of Au-Col-SDF-1a. Indeed, robust standardized assay must be set up for assessment Au-Col-SDF-1a in vivo for elucidation of long-term retention time and interactions with immune microenvironment must be obtained through additional studies in the future.” (Page 23, line 830-835)  

Reviewer 2 Report

It is my great pleasure to have an opportunity to review this manuscript. The authors investigated the efficacy of Au-Col nano delivery system to carry SDF1 alpha for the investigation of biocompatibility and biological performances, particularly the multilineage differentiation capacities of MSCs via upregulated SDF-1α/CXCR pathway, with the enhancement of MMP activities and migration ability. This study is well-written, interesting, and useful contribution for future stem-cell therapies. I have some minor comments and questions that I hope help you revise the manuscript.

1. In the discussion, the authors should emphasize what the Au-Col-SDF1 alpha adds to the subject area compared to other published materials. 2. Have you compared the tissue integrity and biodistribution of Au-Col-SDF-1 alpha to other delivery systems in vivo?

Author Response

Reviewer 2:

It is my great pleasure to have an opportunity to review this manuscript. The authors investigated the efficacy of Au-Col nano delivery system to carry SDF1 alpha for the investigation of biocompatibility and biological performances, particularly the multilineage differentiation capacities of MSCs via upregulated SDF-1α/CXCR pathway, with the enhancement of MMP activities and migration ability. This study is well-written, interesting, and useful contribution for future stem-cell therapies. I have some minor comments and questions that I hope help you revise the manuscript.

  1. In the discussion, the authors should emphasize what the Au-Col-SDF1 alpha adds to the subject area compared to other published materials.

Response:

Thanks for the valuable comment from the reviewer. We have included the description following reviewer’s suggestion in the “Discussion” section “SDF-1α/CXCL12-CXCR4 engagement on DCs are known to promote DCs maturation, survival and migration. In our prior report, we conjugated SDF-1α into Au-collagen nanocarrier and detected their biological effect on dendritic cells [49]. It might provide as a promise approach for specific targeting immunotherapy application. Therefore, compared with this study, Au-Col-SDF-1αnot only stimulate stem cell differentiation by activating the SDF-1α/CXCR4 signaling pathway, but also activate T cells by stimulating dendritic cells, which can be used for immunotherapy, such as the development of novel gold nanocomposite vaccines.” (Page 23, line798-805)  

  1. Have you compared the tissue integrity and biodistribution of Au-Col-SDF-1 alpha to other delivery systems in vivo?

Answer:

Thanks for the suggestion from the reviewer. We have included the description in the “Discussion” section following by reviewer’s suggestion. “The bio-distribution of Au-Col injection into the retroorbital sinus for 24 h. The histological tissues showed the FITC labeled Au-Col could be appeared in most organs after injection, such as the heart, spleen, brain, lung, kidney, and liver was found in our previous literature [22]. Another report s also demonstrated while administered into mice via retro-orbital sinus injection for in vivo assessments. It was also found that in each tissue were obviously observed in fluorescent intensity was highly expression on different organs as well as had a well tissue integrity. Based on these findings, Au-Col nanocarrier may to be safe and showed better retention efficacy in the treatment of mice, which suggests it may have potential in the development of nanodrugs either breast tumor or brain tumor [37, 48].” (Page23, line 821-835)

Reviewer 3 Report

Reviewing report

I.                    "Abstract" and "Introduction" sections :

 English must be improved (see comments in the text).

A paragraph on the physiological roles of SDF-1a and the CXCR4/SDF-1a pathway should be added in the introduction..

II.                  "Materials and Methods" section:

 II.a. Contains inaccuracies and errors i.e. l. 133-134 :  "SDF-1α (10 μg) was purchased from Prospec-Tany Technogene Ltd. (Israel). A 10 ug amount of SDF-1α was mixed with 100 μL of ddH2O to obtain a stock solution of 0.001 μg/μL.

Please see other comments in the text.

II.b. Did you characterized the isolated MSC ?

II.c. Please mention your control condition (0 ppm, no SDF1 and no Col I ) and please be consistent with the abbreviation used for "gold nanoparticles"

II.c. l 300 to 301 : AnnV+/PI- are early apoptotic cells, ANNV-/PI+ are necrotic cells and Ann+/PI+ are late apoptotic and necrotic cells. (NB: necrotic cells may be AnnV +).

II.d. You used different cell densities in your experiments, even when cells were cultured and treated for 3 to 48hours:

-          20,000 cells in 96w (= 62,000 cells/cm2)

-          10,000 cells in 24w (=5,000 cells/cm2)

-          200,000 cells in 6w (=20,000 cells:cm2)

Why ? Don't you think that this may affect the parameters that you evaluate  ?

 II.e. l; 377 : You tested 4 inhibitors associated with cell energy-dependent endocytosis pathways.

Why did you select those 4 inhibitors ?

Are they really "specific" for cellular uptake mechanism inhibition ?

Please explain briefly their mode of action.

II.f. l. 450 : The MTT assay measures cellular metabolic activity ; it is an indicator of cell viability, proliferation and cytotoxicity but not a direct evaluation of those parameters.

Have you ever used another approach to evaluate viability ?

 III.                "Results" section:

III.a. The authors used the student's t-test to perform their statistical analyzes and they compare the experimental conditions vs the control condition (l. 418).

My main concern is that the authors further concluded in the text that significant statistical differences exist between the tested conditions, often claiming that Au-Col-SDF1 is the most effective one (please see comments on l. 452, 474, 480, 512, 514, 521, 542, 577, and so on and in the conclusion section, l. 810-812).

To demonstrate statistical differences between various experimental conditions, the authors should have done ANOVA tests (or adequate non parametric tests if required), eventually followed by post-hoc analysis tests.

To my view, most of the written conclusions are thus inappropriate (see underlined text in Results section).

 Interestingly, the authors conclude in Conclusion section that : " In conclusion, the evidence of good biological performances with biocompatibility elucidates that Au-Col is a promising candidate, while further exploration of endocytosis mechanisms could provide the directions needed for future clinical stem-cell therapies." (l 816-818)

I agree with that comment, but it is not in relation with what is claimed in the result section.

III.b. Western blot data, Fig. 7

Some bands are overexposed.

Were the data normalized for B actin expression (not clear in the M&M section) ? If so, it should be mentioned on the graphs.

IV.                Discussion and Conclusion sections

The discussion consists mainly in the repetition of the results obtained with mention of the conclusions already indicated in the results section. As those conclusions are based on inadequate statistical analyzes, I may not validate them , please see point III.a  above.

My comments are in the text of the manuscript and mentionned in the reviewing report adressed to the authors.

Author Response

Reviewer 3:

  1. “Abstract” and “Introduction” sections: English must be improved (see comments in the text). A paragraph on the physiological roles of SDF-1a and the CXCR4/SDF-1a pathway should be added in the introduction.

Response:

  1. Thanks the valuable comment from the reviewer. We have rewrite and modified the “Abstract” section and “Introduction” section following by reviewer’s kindly suggestion. (marked with blue color)
  2. We have addressed and included the description in the “Introduction” section. “SDF-1a also named chemokine 12 (CXCL12), is expressed in multiple tissues and cell types in which it regulates the homing and function of many stem cells [27], including chemotaxis cytokine receptor-4 (CXCR4 hematopoietic stem or progenitor cells [28,29]” The SDF-1α/CXCR4 axis has been addressed to be associated with cell motility [30]. (Page 3, Line 109-113)

  1. I"Materials and Methods" section:

II.a. Contains inaccuracies and errors i.e. l. 133-134:  "SDF-1α (10 μg) was purchased from Prospec-Tany Technogene Ltd. (Israel). A 10 ug amount of SDF-1α was mixed with 100 μL of ddH2O to obtain a stock solution of 0.001 μg/μL.

Response:

Thanks for the comment from the reviewer. We have modified the sentence and make it to be more easy follow in the “Materials and Methods” section. “Stromal cell derived factor-1a (SDF-1a) was purchased from Israel Prospec-Tany TechnoGene Ltd. A 100 ng/ml of SDF-1a was used as a working concentration in all subsequent experiments.” (Page 3, Line 129-131).

Please see other comments in the text.

II.b. Did you characterized the isolated MSC?

Response:

We have included the characterization data of MSC into the new “Figure S1” section and description in the “M&M” section and “Results” section.

  1. In the “Materials and Methods” section: “To characterize the phenotypes of Wharton’s jelly MSCs, the cells were detached by 2mM EDTA with PBS. The MSCs were washed by PBS containing 2% bovine serum albumin (BSA) and 0.1% sodium azide (Sigma, USA). Next, the cells were cultured with various specific antibodies including CD34-FITC-A, CD45-FITC-A, CD44-PE-A and CD105-PE-A (fluorescein isothiocyanate represented FITC and phycoerythrin denoted as PE). The FITC/PE conjugated IgG1 were applied for isotype controls (BD Pharmingen, USA) by flow cytometer detection.” (Page 4, Line195-202)
  2. In the “Results” section:

“The phenotypes of Wharton’s jelly MSCs used in this study were characterized by CD34, CD44, CD45 and CD105 surface markers. Figure S1 demonstrated the results of each specific marker expression. The expression of CD34 and CD45 endothelial markers was analyzed as 1.87 % and 0.80% (negative markers). The cells expressed CD44 and CD105 positive markers as 99.7 % and 99.6 %, which demonstrated the expression of MSC surface markers by flow cytometry analysis.” (Page 10, Line 454-459)

II.c. Please mention your control condition (0 ppm, no SDF1 and no Col I) and please be consistent with the abbreviation used for "gold nanoparticles"

Response:

  1. Thanks for the valuable comment from reviewer. We have modified the original description of “containing Au, SDF-1α, Au-Col and Au-Col-SDF-1α” into “containing control (without treatment), Au, SDF-1α, Au-Col and Au-Col-SDF-1α.”
  2. We have modified the abbreviation of “gold nanoparticle” into “Au”.

II.c. l 300 to 301: AnnV+/PI- are early apoptotic cells, ANNV-/PI+ are necrotic cells and Ann+/PI+ are late apoptotic and necrotic cells. (NB: necrotic cells may be AnnV +).

Response:

We thank the valuable comment from the reviewer. We have modified the description in the “Results” section “Annexin V+/PIpopulation was represented as apoptotic cells, Annexin V/PI+ population was necrotic cells and the Annexin V+/PI+ population was dead cells.” into “Annexin V+/PIpopulation was represented as early apoptotic cells, Annexin V/PI+ population was necrotic cells and the Annexin V+/PI+ population was late apoptotic and necrotic cells.” (Page 6, Line 301-304)

II.d. You used different cell densities in your experiments, even when cells were cultured and treated for 3 to 48hours:

-     20,000 cells in 96w (= 62,000 cells/cm2)

-     10,000 cells in 24w (=5,000 cells/cm2)

-     200,000 cells in 6w (=20,000 cells:cm2)

Why? Don't you think that this may affect the parameters that you evaluate?

Response:

  1. We agree with the comment from the reviewer that different cell densities will affect the distribution of cell numbers due to the either difference surface area or different time point. All experimental conditions and procedures are carried out with reference to standard published journals. Therefore, all experimental conditions, cell numbers, and culture conditions are referenced and operated according to standard experimental procedures.
  2. To avoid the misunderstanding, we also modified the better description in the “Materials and Methods” section as list below:
  • “MSCs (2´104) were loaded were seeded in a 96-well culture plate.” (Page 5, Line 204)
  • “MSCs (2´105) were loaded were cultured in 6-well culture plates.” (Page 5, Line 218-219)
  • “MSCs (2´105) were loaded were seeded in a 6-well culture plate.” (Page5, Line 229-230)
  • “MSCs (1´104) were incubated.” (Page 5, Line 240)
  • “MSCs (1´104) were cultured in 24-well culture plates.” (Page 5, Line 252-253)
  • “MSCs (2´105) were seeded in 6-well culture plates.” (Page 6, Line 273)
  • “MSCs (2´105 ) were loaded in 6-well culture plate.” (Page 6, Line 283)
  • “MSCs (2´105) were loaded in 6-well culture plate.” (Page 6, Line 295)
  • “MSCs (2´105) were seeded in 10 cm2 culture dishes.” (Page 7, Line 310)
  • “MSCs (1´104) were loaded in a 24-well culture plate.” (Page 7, Line 329)
  • “MSCs (2´105) were seeded in 10 cm2 culture dishes.” (Page 7, Line 354)
  • “MSCs (1´104) were loaded into 24-well culture plates.” (Page 8, Line 367)
  • “MSCs (1´104 ) were seeded in 24-well culture plates” (Page 8, Line 395-396)

 II.e. l; 377: You tested 4 inhibitors associated with cell energy-dependent endocytosis pathways. Why did you select those 4 inhibitors? Are they really "specific" for cellular uptake mechanism inhibition? Please explain briefly their mode of action.

Response:

Thanks for the valuable comments from the reviewer. We have been addressed the mode of action in the original “Discussion” section (Page 23, Line 806-817)

II.f. l. 450: The MTT assay measures cellular metabolic activity; it is an indicator of cell viability, proliferation and cytotoxicity but not a direct evaluation of those parameters. Have you ever used another approach to evaluate viability?

Response:

Thanks the valuable comments from the reviewer.

  1. MTT is currently in scientific research and is well documented as a favored measurement reagent for testing ell viability as well as cell growth efficiency. Therefore, in this paper we use MTT to detect cell viability with reference to previously published literatures.
  2. At the same time, we also detected the cell cycle by flow cytometry and found that Au-Col-SDF-1a can indeed increase the S-phase cycle, thus further confirming that Au-Col-SDF-1a can indeed induce cell proliferation and growth (Figure 5A).
  3. At the same time, in the results of western blot, it was also found that Au-Col-SDF-1a can also increase the expression of the cell cycle-promoting cell cycle regulatory factor of Cyclin D1 (Figure 7).
  4. Therefore, according to the research results either Figure 5A or Figure 7, it was demonstrated the Au-Col-SDF-1a can promote the better cell growth capacity in this study.  

 III.  "Results" section:

 III.a.

  1. The authors used the student's t-test to perform their statistical analyzes and they compare the experimental conditions vs the control condition (l. 418).

Response:

Thanks for the valuable comment from the reviewer. We apology for this typo mistake. We have corrected and made the new description in the “Materials and Methods” section. “Statistical analysis was performed by SPSS software (version 17.0). Differences between mean values were determined by a one-way ANOVA followed by a Bonferroni’s test. Probability values (p)<0.05 were regarded to have significant differences between treatments.” (Page9, Line 420-423)

  1. My main concern is that the authors further concluded in the text that significant statistical differences exist between the tested conditions, often claiming that Au-Col-SDF1is the most effective one (please see comments on l. 452, 474, 480, 512, 514, 521, 542, 577, and so on and in the conclusion section, l. 810-812).

To demonstrate statistical differences between various experimental conditions, the authors should have done ANOVA tests (or adequate non parametric tests if required), eventually followed by post-hoc analysis tests.

Response:

Thanks for the valuable comment from the reviewer. We apology for this typo. We have corrected and made the new description in the “Materials and Methods” section. “Statistical analysis was performed by SPSS software (version 17.0). Differences between mean values were determined by a one-way ANOVA followed by a Bonferroni’s test. Probability values (p)<0.05 were regarded to have significant differences between treatments.” (Page9, Line 420-423)

  1. To my view, most of the written conclusions are thus inappropriate (see underlined text in Results section).

 Interestingly, the authors conclude in Conclusion section that: " In conclusion, the evidence of good biological performances with biocompatibility elucidates that Au-Col is a promising candidate, while further exploration of endocytosis mechanisms could provide the directions needed for future clinical stem-cell therapies." (l 816-818)

I agree with that comment, but it is not in relation with what is claimed in the result section.

Response: 

Thank the valuable comment from the reviewer. We have corrected the typo of “Au-Col” and modified into “Au-Col-SDF-1a”. (Page 24, Line 849)

III.b.

  1. Western blot data, Fig. 7 Some bands are overexposed.

Response:

Thanks for the valuable comment from reviewer. We have following and adjust the background to be dark.

Were the data normalized for B actin expression (not clear in the M&M section)? If so, it should be mentioned on the graphs.

  1. Were the data normalized for b-actin expression (not clear in the M&M section)? If so, it should be mentioned on the graphs.

Response:

Thanks the valuable comment from the reviewer. We have include the more detail description in the “Materials and Methods” section “b-actin was used as a loading control, to normalize total protein amounts and check for eventual protein degradation in the samples.” (Page 7, Line 325-327)

  1. Discussion and Conclusion sections

The discussion consists mainly in the repetition of the results obtained with mention of the conclusions already indicated in the results section. As those conclusions are based on inadequate statistical analyzes, I may not validate them, please see point III.a above.

Response:

We appreciate the comment and apology for causing misreading from this manuscript. The statistical result is a typo. All research results are completed using the AVONA statistical method. We have corrected all statements and results in the “Materials and Methods” section. (Page 9, Line 420-423)

Comments on the Quality of English Language

My comments are in the text of the manuscript and mentionned in the reviewing report adressed to the authors.

Response:

Thanks for the valuable comments from the reviewer. We have carefully and modified the quality of English editing in the “Abstract” section and “Introduction” section following by reviewer’s suggestion.
